# COVID-19 and Prosthetic Emergencies, Home Care in Fragile Patients: A Case Report

**DOI:** 10.3390/healthcare10081407

**Published:** 2022-07-27

**Authors:** Saverio Ceraulo, Paolo Caccianiga, Carmelo Casto, Marco Baldoni, Gianluigi Caccianiga

**Affiliations:** 1School of Medicine and Surgery, University of Milano-Bicocca, 20900 Monza, Italy; saverio.ceraulo@unimib.it (S.C.); marco.baldoni@unimib.it (M.B.); gianluigi.caccianiga@unimib.it (G.C.); 2Independent Researcher, 73000 Lecce, Italy; dottorecasto@gmail.com

**Keywords:** COVID-19, prosthesis, fracture, resin, home care

## Abstract

A case of home care is proposed on a frail non-ambulatory patient who presents an old lower total prosthesis in resin broken in several parts. The various pieces of the prosthesis were joined by the patient, as if it were a puzzle, using a glue for plastics and wood. The union of the parts attached with glue was the consequence of the lockdown in the COVID-19 period and of the economic hardship experienced by the elderly disabled patient during the pandemic period. The procedure for preserving the glued parts was carried out carefully, trying not to modify the edges of the glued pieces, to join them correctly, thereby restoring the correct occlusion to subsequently perform the relining. The old lower total prosthesis obtained after the repair and relining operations allowed for the restoration of the patient’s chewing and smile. The procedure presented is easily repeatable, risk-free and achievable even in a short time, satisfying elderly non-self-sufficient patients who need interventions for prosthetic emergencies during a period of confinement.

## 1. Introduction

The coronavirus (COVID-19) pandemic has upset the habits of different populations and the second wave highlighted even more in some people the fear of being infected. In some fragile and vulnerable individuals, governments’ preventative strategies to protect and promote health also created adverse effects when exposed to long-term news relating to the pandemic [1]. In the period of the COVID-19 pandemic, appeals were launched not to neglect the continuity of care for patients suffering from chronic diseases, with the aim of avoiding delays in treatment as much as possible [2]. The COVID-19 pandemic effect has led to the development of greater protection for those fragile subjects who, unable to move, even if accompanied, tried to wait out the need for treatment of prosthetic dental problems, fearing a danger of getting infected. Often, the prostheses that fracture easily due to the reduced thickness of the resin are the resin partial prostheses mainly used as provisional measures in partially edentulous subjects. As well as being easier to break, they are also easier to repair than a skeletonized prosthesis [3]. A skeletonized prosthesis is recommended because it is aesthetically valid and to obviate the disadvantage of breaking the prosthesis; and, from the point of view of satisfaction, removable metal prostheses have a higher degree of approval in partially edentulous patients [4]. In the case of upper and lower total prosthesis, the breakage event is more evident for the lower prosthesis as its horseshoe design creates greater tension and, therefore, fragility in the center of the prosthesis in the fifth sextant. The repair of an acrylic resin prosthesis is not difficult, but some indications must be respected regarding the preparation of the surfaces to be repaired, to avoid further stresses to the resin [5,6,7,8,9], and the type of polymerization used must have good mechanical resistance characteristics [10]. Some complete denture wearers report difficulty with daily activities, especially activities related to denture retention and stability. A study by Limpuangthip et al. [11] focused on the importance of stability and retention of a complete denture. The purpose of their clinical study was to determine the association between professionally-based assessment of complete denture quality and multiple patient-based outcomes: oral health-related quality of life (OHRQoL), eating satisfaction and masticatory performance. Denture stability showed a stronger association than retention. They concluded that retention and stability were important indicators in estimating the masticatory ability and OHRQoL of complete denture wearers. It is also necessary to point out the importance of the tongue activity, which can affect the stability of removable mandibular dentures. Żmudzki et al. [12] tried to demonstrate that tongue force improves mandibular complete denture stabilization on the atrophied foundation during mastication load transfer with a typical balanced occlusion. They observed that, despite the lack of substantial improvement in stability with the addition of tongue forces, tongue action that contributes to a slight reduction in sliding may help reduce common frictional trauma resulting from cyclic movement during chewing.

In the case described below, the 77-year-old disabled patient with a history of tracheostomy for laryngeal carcinoma, hypertension and diabetes, for fear of viral infection, in conditions of necessity for fracture of the lower prosthesis, repaired the prosthesis using a glue for plastics, synthetic resins, panels of different materials and wood. As the surfaces of the bonded resin parts were rough, because they were impregnated with glue, this created real retentions of dental plaque over the days, generating halitosis [13]. The patient’s only relief in reducing halitosis was the use of a chlorhexidine-based mouthwash [14]. The patient’s increasingly worsening discomfort led family members to find a solution by looking for a dentist who could solve the problem without them having to leave home.

## 2. Materials and Methods

In the period of the pandemic, many patients experience the discomfort of having to adapt, also resorting to means and materials available in homes to repair a broken prosthesis, thus avoiding the possible risk of contagion with the COVID-19 virus and psychological discomfort associated with going to the dentist. The patient in question, through family members, contacted us out of desperation, describing the reason for the call as follows: after having repaired the lower prosthesis several times and having convinced himself that he could no longer continue chewing, he realized that the upper prosthesis was also broken. The following materials and equipment were prepared for the repair and correction of the prostheses: two scalpels, red wax, pink cold acrylic resin, polysulfide for condensation and laboratory motor with portable dental unit burs. After a first telephone triage and a subsequent one before entering the patient’s home, the basal temperature was measured for the patient and family members as a precaution, while the medical staff wore personal protective equipment before entering the patient’s home [15]. Before starting the work, anamnestic data were collected, and during this phase, the patient reported that the lower prosthesis had broken and he had repaired it several times with a glue for plastic resins and wood. During the interview, the patient showed difficulty in speaking, given the tracheostomy operation, and reported incessant discomfort from food accumulation during chewing in the blocked parts, along with bad breath [16]. At the intraoral examination, total upper edentulism was noted below only two root residues 43 and 44. The lower prosthesis showed some fractures in different parts assembled incorrectly using a glue for plastic resins and wood, found in the home, as shown in Figure 1. The first phase was to evaluate the occlusion, trying to understand how it was possible that the prosthesis could remain in equilibrium considering that the glued parts had displaced all the occlusion between the two arches on the right side of a tooth, as shown in Figure 2. Before proceeding with the repair of the prosthesis, we removed the glue with the aid of the scalpels and repositioned the broken parts, trying not to alter the edges of the individual pieces adhering to the resin. The second phase was to combine the pieces of the clean resin of the glue and assemble them correctly by blocking them first with wax and then with cold acrylic resin, to allow an evaluation in the mouth of the correct occlusion of the patient. After evaluating the correct occlusion and position in the oral cavity, we moved on to the third phase, that is, the impression with the polysulfide, to perform the repair and reline both the lower and upper prostheses, as shown in Figure 3. The last phase, the delivery of the two prostheses and occlusal check, was performed on the same day, as shown in Figure 4.

## 3. Results

The acrylic resin for prosthetic bases has the advantage of being easily repairable. After meticulously removing the glue that joined the resin pieces using manual and rotating tools, without compromising the edges, this allowed us to compose the clean parts like a puzzle, to perform the repair and relining of the lower prosthesis. We developed the two prostheses in the correct occlusion for the patient in just two sessions on the same day.

## 4. Discussion

The difficulty in getting fragile, non-self-sufficient or non-ambulatory patients to go to dental health facilities is increasing, especially with the aging of the population, but even more so with the COVID-19 pandemic event, with many patients still worried about contagion. The possibility of treating fragile patients in their own homes with certified portable materials and equipment allows the dentist to reach those people who would otherwise be forced to call medical vehicles several times for transport to health facilities, resulting in inconvenience and increased costs. The health professionals who have more contact with frail elderly patients with dental prostheses are general practitioners. It would be desirable in the future to introduce a questionnaire, to be distributed to all general practitioners, to be filled out by patients or family members of those who undergo prosthetic prophylaxis. With the aim of managing and minimizing complications due to mobile prostheses in fragile patients, the practitioner must educate the patient on the importance of dental adhesives in complete denture wearers, to avoid problems of retention and stability and to promote chewing ability and quality of life. A study by Mendes et al. [17] focused on this matter and demonstrated that the original brands of DA have a significantly higher retentive ability than the white brands.

## 5. Conclusions

The period of confinement experienced by many frail elderly people during the pandemic has led to some developing the art of fending for themselves. During the pandemic, it has often happened that we have seen resin prostheses repaired with home methods, with the consequent need for us to construct a new prosthesis due to the impossibility of repairing the old prosthesis. Breaking this news, for some elderly subjects, creates a sense of guilt. The patient we treated, having repaired the prosthesis several times, realizing that it would have created more damage to try to do so again, called in help from family members to definitively solve a problem that, for a fragile person, would seriously affect their normal daily life and psychological balance to resolve. The smile and the satisfaction of the patient, when wearing his old but now repaired and relined prostheses, enabled him to forget at that moment his fear of COVID-19 and the discomfort experienced during chewing. Plus, in addition to the economic savings made by avoiding the need for the construction of new prostheses, it benefitted the patient psychologically to think that the prostheses were the same as before and he had not destroyed them with his handiwork.

## Figures and Tables

**Figure 1 healthcare-10-01407-f001:**
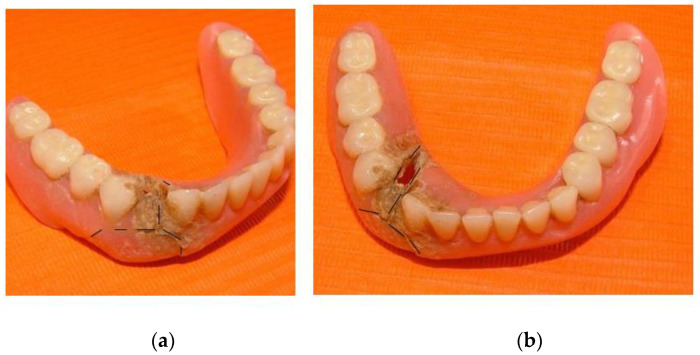
Prosthesis repaired by the patient: (**a**) lateral view; (**b**) frontal view.

**Figure 2 healthcare-10-01407-f002:**
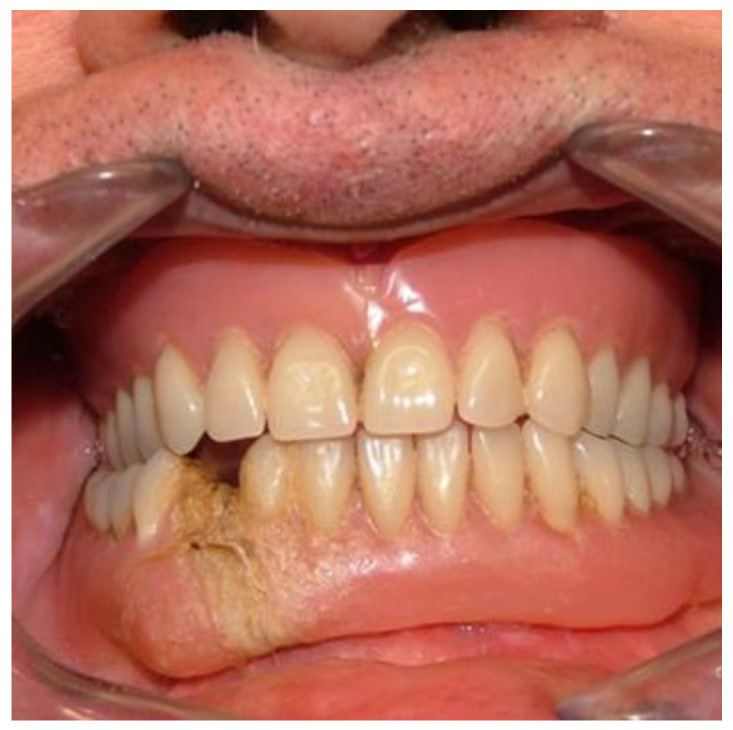
Intraoral view of the prosthesis repaired by the patient.

**Figure 3 healthcare-10-01407-f003:**
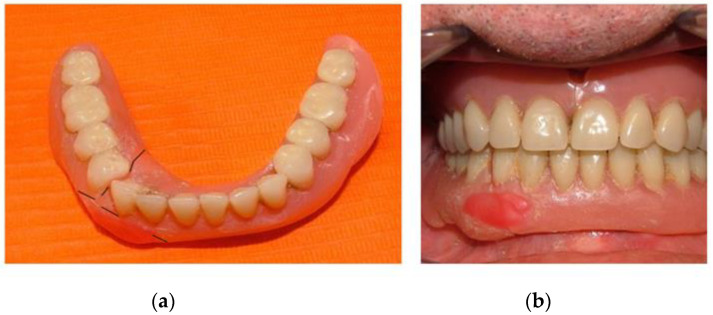
(**a**) Patient’s prosthesis reassembly; (**b**) evaluation of the occlusion, in which the fracture of the upper prosthesis is noted.

**Figure 4 healthcare-10-01407-f004:**
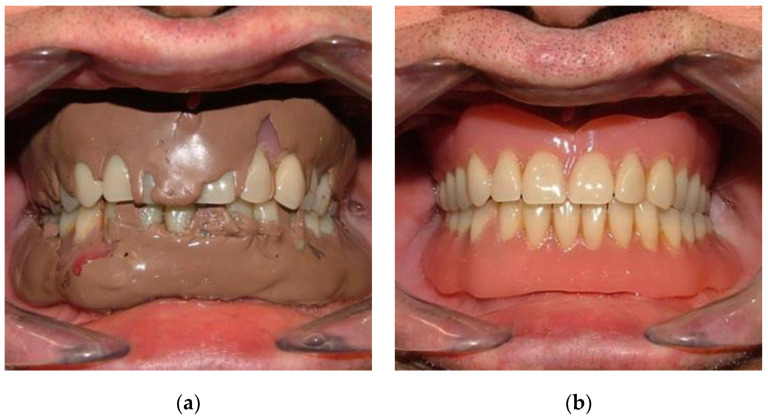
(**a**) Impression with polysulfide to repair and reline; (**b**) prosthesis repaired and relined.

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
