# Peer review of "COVID-19 and Prosthetic Emergencies, Home Care in Fragile Patients: A Case Report"

_healthcare, 2022, doi:10.3390/healthcare10081407_

Round 1

Reviewer 1 Report

Firstly, thank you for opportunity to review very interested article. I don't feel qualified to judge about the English language and style due to not native language.

1. The title reflect the main subject about COVID and care of affect patients with dental problems, title was clear and easy to understand.

2. The abstract summarize and reflect the work described in the manuscript.

3. The key words reflect the focus of the manuscript.

4. The manuscript adequately describe the background, present status, and significance of the study. The authors explain COVID related health policy effect especially in bed ridden patients.

5. The manuscript describe methods in adequate detail, study subjects were clear, with demonstrate IRB number or text to human ethics consideration.

6. The research objectives achieved by the experiments used in this study.

7. The manuscript interpret the findings adequately and appropriately, highlighting the key points concisely, clearly, and logically. However, in discussion part I suggested the authors to discuss about how to manage in case with this problems.

8. Tables and figures sufficient, good quality and appropriately illustrative of the paper contents.

9. The manuscript meet the requirements of biostatistics.

10. The manuscript cite appropriately the latest, important, and authoritative references in the introduction and discussion sections. However, some of references were incorrect style for this journal.

Author Response

Dear reviewer,

Thank you for your kind and constructive review.

We have revised the manuscript according to your suggestions, including the implementation of the discussion and the correction of receferences' style.

Best regards,

Paolo Caccianiga

Reviewer 2 Report

The authors present a case report with a case of home care in the Covid-19 period, they describe a treatment of a fracture of a total prosthesis in resin broken in several parts. It is an interesting and relevant article, and it structure seems adequate. 

Although the study has important clinical and scientific implications, the article needs a discussion and introduction with more evidence on the topic and more articles to compare. 

The bibliographic references are insufficient, some corrections should be mentioned:  

- In the introduction, a paragraph should be introduced about the importance of stability and retention of a complete denture, this because the authors describe that it was necessary to perform a relining. I suggest the article: “Limpuangthip N, Somkotra T, Arksornnukit M. Modified retention and stability criteria for complete denture wearers: A risk assessment tool for impaired masticatory ability and oral health-related quality of life. J Prosthet Dent. 2018 Jul;120(1):43-49. doi: 10.1016/j.prosdent.2017.09.010. Epub 2017 Nov 29. PMID: 29195820.” 

Furthermore, in the introduction, must mention the importance of the tongue activity, involving can affect the stability of removable mandibular dentures. I suggest the article: 

“Å»mudzki J, Chladek G, Krawczyk C. Relevance of Tongue Force on Mandibular Denture Stabilization during Mastication. J Prosthodont. 2019 Jan;28(1):e27-e33. doi: 10.1111/jopr.12719. Epub 2017 Dec 29. PMID: 29285830.2 

- In the discussion, the authors should mention the importance of dental adhesives in complete denture wearers in order to avoid these problems (retention and stability). I suggest the article: “Mendes J, Mendes JM, Barreiros P, Aroso C, Silva AS. Retention Capacity of Original Denture Adhesives and White Brands for Conventional Complete Dentures: An In Vitro Study. Polymers (Basel). 2022 Apr 26;14(9):1749. doi: 10.3390/polym14091749. PMID: 35566919; PMCID: PMC9104604.” 

Author Response

Dear reviewer,

Thank you for your kind and constructive review.

We have revised the manuscript according to your suggestions, including the implementation of the introduction and discussion, by citing the articles you suggested.

Best regards,

Paolo Caccianiga

Round 2

Reviewer 2 Report

Dear authors
Thanks for the quick corrections made to the article.
The submitted corrections to the article "COVID-19 and prosthetic emergencies, home care in fragile patients: a case report" significantly improved the document. The article has conditions to be published.

Best regards